# Response of Chironomids to Key Environmental Factors: Perspective for Biomonitoring

**DOI:** 10.3390/insects13100911

**Published:** 2022-10-07

**Authors:** Bruno Rossaro, Laura Marziali, Angela Boggero

**Affiliations:** 1Dipartimento Ambiente Territorio, Università degli Studi di Milano, Via Celoria 2, 20133 Milano, Italy; 2National Research Council—Water Research Institute (CNR-IRSA), Via del Mulino 19, 20861 Brugherio, Italy; 3National Research Council—Water Research Institute (CNR-IRSA), Corso Tonolli 50, 28922 Verbania Pallanza, Italy

**Keywords:** chironomidae, freshwaters, macroinvertebrates, ecological indicators

## Abstract

**Simple Summary:**

Benthic macroinvertebrates of inland waters, including running waters and lakes, are frequently used in biomonitoring. Sometimes, environmental data associated with species lists are not available; in this situation traits or functional adaptations of species to environment can be considered as a tool to translate the list of species into a useful index to evaluate the environmental quality a body of water.

**Abstract:**

Chironomids are the species-richest family among macroinvertebrates and are often used as indicators of ecological conditions in inland waters. High taxonomic expertise is needed for identification and new species are still being described even in the well-known West Palearctic region. Our Microsoft Access relational database comprises data on Chironomid species collected in rivers and lakes in Italy and some other European countries over a period of about 50 years, often associated with physical-chemical data, but in some cases, only data on Chironomids are available with no associated environmental data. The aim of the present paper was to propose the calculation of ecological traits of Chironomid species as a tool to derive information on water quality, when only data on Chironomid species composition are present, while environmental data are lacking. Traits summarizing the species’ response to environmental variables were evaluated, with emphasis on natural and man-influenced factors: current velocity, water temperature, conductivity, dissolved oxygen, and nutrients. Traits calculations were carried out in the R environment using a subset of our data, including both environmental data and Chironomid abundances. The relations between sites, Chironomid, species and traits were evaluated using correspondence analysis and other multivariate methods. The response of species showed an interaction among different factors, with the possibility of ordering species along a single environmental gradient, extending from cold running waters to warm standing waters, with few exceptions.

## 1. Introduction

The analysis of environmental factors responsible for macroinvertebrate assemblage structure has a long history. Among macroinvertebrates, Chironomids are considered a hard-to-identify group, therefore studies concerning macroinvertebrates as bioindicators have been often limited to Ephemeroptera, Plecoptera, and Trichoptera (EPT) [1]. Taxonomic problems were brought forward to justify this choice, but even though Chironomids are a difficult group, this is not a valid reason to disregard them. Chironomids include species living in almost all water bodies, sometimes present with a very large number of species, so their exclusion can lead to a serious misjudgment in water quality assessment [2].

A frequently overlooked problem in the development of a biotic index is species identification accuracy. Particularly in taxonomic hard groups, frequent mistakes in species identification were observed. It must be pointed out that different species within the same genus may show a different indicator value, therefore, an index based only on genus identification can lead to misleading conclusions with respect to an index based on the identification of species [3]. However, an intermediate level between genus and species, which we termed “morphotaxon”, can be used to describe the ecological responses of Chironomidae [4].

It is well known that different Chironomid species colonize different river reaches and lake types, suggesting the existence of krenal, rhithral, and potamal species in running waters, and littoral, sublittoral and profundal species in lakes [2,4]. In fact, species distribution can be easily related to a few environmental factors such as substrate type, habitat heterogeneity or alteration [5,6,7], distance from the source or mouth [8], current velocity [9], water temperature [5,10], lake depth [11,12,13], conductivity [2,4], salinity [5], oxygen content [5,14,15], pH [11], water quality [6], submerged plants [11], sediments organic matter content [10]. This result was evident in running waters just one century ago, with Orthocladiini and Tanytarsini dominating the upper reaches of rivers and Chironomini in the lower reaches. A similar separation of tribes was observed in lakes, leading to the separation of oligotrophic *Orthocladius/Tanytarsus* lakes as opposed to eutrophic *Chironomus* lakes [2,5,16].

On the other hand, the attempt to use Chironomid species as indicators of toxic substances did not make much progress, with the same tolerant/intolerant species probably being tolerant/intolerant to a set of many other different factors [17]. In contrast, studies concerning the response of Chironomid species to habitat heterogeneity or alteration were more fruitful [5,6,7].

The use of species identification in the assessment of water quality was criticized and then refined considering biological and ecological traits [18,19,20,21,22,23], suggesting that non-taxonomic aggregation of taxa as similar as possible in their species traits could aid in the interpretation of information given by a taxonomic list of species. For example, biological traits were preferable to taxonomic species lists in analyzing the response of multiple stressors in central European lowland rivers [24].

The problem is that the possibility of translating a list of species into biological and ecological traits needs basic research to prepare such a translation.

The aim of the present paper is to discuss the advantages and limitations of the use of ecological traits with respect to the taxonomic approach, testing a large database, including both lotic and lentic habitats, with multivariate data analysis. The discussion considers the situations where environmental data associated with species lists are lacking or scanty, and trait calculation is proposed as a method of overcoming the problem of missing environmental data.

## 2. Materials and Methods

### 2.1. Chironomid Database

A large database including data on Chironomid species collected in rivers and lakes in Italy and in some other European countries over a period of about 50 years was considered. Physical-chemical data associated with Chironomid samples were available, but only for a subset of samplings. Data were filed in a relational database in Microsoft Access© which is the property of the corresponding author. An extract of these databases is available in Appendix A. Data were stored in different Tables; the description of these Tables is here summarized.

Appendix A. **1 Species**: it contains a list of the variables used, including both environmental variables (morphometric, physical, chemical) and species belonging to the family Chironomidae; species were aggregated in species groups (morphotypes), each morphotype corresponding to a genus, a subgenus, a species group or single species [4]. As regard species, the table contains the species name, author, year of the original description, and taxonomic status (senior synonym, junior synonym, new combination) as additional fields. 

Appendix A. **2 Sites**: it contains a list of the sampling localities, and other additional fields such as latitude, longitude, altitude, distance from the source (for running waters), depth (for lakes), habitat (krenal, kryal, rhithral, potamal, littoral, sublittoral, profundal, etc.).

Appendix A. **3 Conn**: it connects each environmental variable or species with the sampling station and a numerical value; for environmental variables, this number is the measured value, for species it is an index of abundance (see below); additional fields are sampling year, month, day, sampling tool, and bibliographic source of information (when appropriate).

The samples here selected for data analysis included larvae collected with different tools, such as Surber net, kick net, hand net, grab, etc., and measures of environmental variables (e.g., water temperature, conductivity, nutrients, etc.) associated with Chironomid samples, when available. The species abundance value was the total number of specimens in the sample, after sorting under a stereomicroscope LEICA MZ12.5 magnification. However, since data on Chironomid larvae were mostly derived from quantitative or semiquantitative sampling techniques [25,26] for calculations species abundance was expressed as a total number of individuals per m^2^.

A crosstab query was then created with sites and other variables describing the sampling site as rows, and environmental variables or species as columns.

The crosstab query created produced a matrix with 9127 sampling sites, including lentic and lotic waters, sampled in different years and months, in Italy above all [4,27], but including also data from Algeria [28] and other countries in Europe [6]. A complete list of the sites examined is reported in Appendix A. The same query included **160** columns, that is a row label, a sequence number, **6** factors, **11** environmental variables, and **143** Chironomid taxa. The **11** environmental variables included were: sampling year, sampling month, altitude in m a.s.l., distance from the source in km, O_2_ content in mg L^−1^, conductivity in µS cm^−1^, pH, total phosphorous in µg l^−1^, N-NO_3_ in mg L^−1^, N-NH_4_ in µg l^−1^, water temperature in °C. The **4** factors were: habitat, river basin, water body, and sampling station (Appendix A).

The sampling year was included to detect potential temporal trends of species during the long time period considered (about 50 years). Month was used to indicate seasons. 

The taxa included in the analysis were the morphotypes or species groups described in [3]; in the following section of the present work these taxa will be named “species” for simplicity, even if they are often taxa larger than species (i.e., genus or group of species).

The sites where less than 5 species were present and species present in less than 50 sites were excluded, leaving a matrix with **91** species in **2258** sites aggregated in 10 different habitats: glacial streams (K = kryal), springs (S = krenal), rhithral streams (R = rhithral), lowland rivers (P = potamal), alpine lakes (ALA), lowland large lakes (LL), small lakes (LS), volcanic lakes (V), Mediterranean lakes (ME) and brackish waters (B). These 10 habitats were further divided into 102 waterbodies. The delimitation of these habitats is described in other publications [3,27,29,30].

### 2.2. Data Analysis

The crosstab query generated a matrix with n sites as rows and p species + s environmental variables as columns (**_n_M_p+s_**), which was input into an R script (Appendix A). The **M** matrix was separated into an **_n_L_p_** matrix of species and an **_n_R_s_** matrix of environmental variables. Each environmental variable was used to calculate: 1- a correlation matrix between each species and the environmental variables **_p_C_s_**; 2- a weighted mean of each environmental variable for each species, i.e., means of each environmental variable weighted according to species abundances, which can be considered the optimum for each species; 3- a weighted standard deviation, which can be considered a measure of species tolerance. The weighted mean of each environmental variable for each species generated a trait matrix **_p_U_s_** with p species as rows and s environmental variables as columns [4,31]. The presence of missing data in the **_n_R_s_** matrix forced us to calculate matrices **_p_C_s_** and **_p_U_s_** matrices using only the available data.

The **_n_L_p_** matrix, including the reduced n (=2258) sites and p (=91) species, and the **_p_U_s_** matrix, including the same species and s (=11) traits, were analyzed with a correspondence analysis (CA = unconstrained ordination) [31,32]. The **_n_L_p_** values were log(x + 1) transformed before calculation. As a second step, a canonical constrained ordination was carried out using the transpose of **_n_L_p_**, that is **_p_L’_n_**, and **_p_U_s_** as input matrices. As the last step, the **_n_L_p_** matrix was post multiplied by the **_p_U_s_** matrix, submitted an unconstrained ordination and compared with the previous results. The large amount of missing data in the **_n_R_s_** matrix hindered the ability to carry out a canonical constrained ordination between the **_n_L_p_** and **_n_R_s_** matrices. 

The **sites × species** matrix **_n_L_p_** was post-multiplied by **species × traits _p_U_s_** matrix to obtain a **site × traits** matrix (**_n_L _p_U_s_**), i.e., a matrix with sites as rows and species traits as columns. This **_n_L _p_U_s_** matrix was also submitted to correspondence analysis. 

A discriminant analysis was carried out to test the goodness of classification in different habitats using the Chironomid taxa assemblages: both the **_n_L_p_** and the **_n_L _p_U_s_** matrices were submitted to multiple discriminant analysis, using the habitats as a grouping factor. Finally, a cluster analysis of the site × species matrix was carried out using the complete linkage clustering method [32] to detect clusters of species.

**Software**: All data analyses were carried out in the R environment [33], using the packages vegan, Hmisc, mass, and scatterplot3D [34], including R scripts produced by the corresponding author, which are available on request.

## 3. Results

Measures of the 11 environmental variables were available for a reduced number of sites (Table 1), so the correlations, weighted means, and standard deviations of each environmental variable with each species were calculated using sites where both species and environmental data values were available (see Methods); when less than 4 records were available for the couple environmental variable-species, correlations were not calculated and mean values and standard deviations of the environmental variable calculated over all the other species were assigned to these species.

Highly significant correlations (*p* < 0.01) between species abundance and environmental variables were observed for a reduced number of species (Table 2 and Appendix A).

The weighted means, considered the optimum values for each species [29,31], were used to create a **species**
**× traits** matrix **_p_U_s_** with p species as rows and s environmental variables as columns (Table 3). Weighted standard deviation and the number of observations available are reported in Appendix A.

The **sites × species** matrix **_n_L_p_** was submitted to a correspondence analysis; three major gradients (Figure 1, Figure 2, Appendix A and Appendix A, and Appendix A) were evidenced, the former accounting for 7.6% of the total variance, the second 5.1%, the third 3.8% of the total variance, with eigenvalues equal to 0.71, 0.48, and 0.35, respectively. The species and sites ordered in the plane of the two axes showed the typical horseshow or arch effect [31,32]. The first gradient separated running waters from standing waters, the second separated upstream stations from downstream stations in running waters, with the following sequence (Figure 1, Figure 2, Appendix A and Appendix A): 1- frigo-stenothermal species living in kryal were plotted in the bottom left of the graph; 2- rhithral species living in streams were plotted above the former; 3- eurithermal species, living in potamal, were plotted at the apex of the arch, extending from the top to the right part of the plot; 4- species living preferably in lentic waters, were plotted on the right part of the graph; 5- species living in springs were plotted in the central part of the area. Further separations of species from small alpine lakes, such as *Paratanytarsus austriacus*, *Heterotrissocladius*, *Corynoneura*, and *Zavrelimyia*, were plotted in the center of the area, species characterizing profundal zone of lowland large lakes, such as *Micropsectra radialis* and *Paracladopelma*, were also plotted closer to the center of the area at the right of alpine lakes species, while small prealpine and volcanic lakes were grouped on the right of the plot. This separation was still better emphasized in a 3D plot (Figure 1 and Appendix A), where kryal, rhithral, krenal, potamal and lentic species were separated.

A polynomial of the second degree was fitted to species scores of the two first axes (Figure 2), resulting in a multiple R-squared of 0.6845, and an adjusted R-squared of 0.6773 (F-statistic = 95.47 with 2 and 88 degrees of freedom, *p*-value = 2.2 ∗ 10^−16^, residual standard error = 0.7344 with 88 degrees of freedom). The species more distant from the parabolic curve are visible in Figure 2 and are also evident in Appendix A, where all species names are plotted. Species from small Alpine lakes and from profundal zones of large lakes are the ones most deviating from the parabolic curve.

The environmental variables were included as passive variables in the map and were converted into factors with 6 different levels. When missing data were present, a level, plotted as void circles, grouped these data. The factors included were habitat, station (Figure 3), altitude, distance from the source (Figure 4), temperature, conductivity (Figure 5), oxygen, total phosphorous (Figure 6), nitrate, and ammonium nitrogen (Figure 7). 

The **species × traits** matrix **_p_U_s_** was also subjected to a correspondence analysis (Figure 8, Appendix A). The first 2 axes accounted for 70% and 21% of the total variance. Eigenvalues were 0.14 and 0.04, respectively. The first gradient separated species according to an upstream-downstream gradient, with the extreme scores assigned to altitude, distance from the source and conductivity. The second gradient separated species according to a trophic gradient, with the extreme scores assigned to oxygen, and N-NH_4_. *Tanypus* and *Chironomus riparius* (=*thummi*) were plotted in the bottom left area, as well as other tolerant species, while *Diamesa* species were plotted in the bottom right area. Species requiring high O_2_ content such as *Paralauterborniella*, *Pagastiella* and *Stempellina* were plotted at the top of the graph.

The **sites × species** matrix was transposed (**_p_L’_n_**) and a canonical constrained ordination (CCA) was carried out relating this matrix with the species × traits matrix **_p_U_s_** (**_p_L’_n_**~**_p_U_s_**) (Figure 9 and Appendix A, Appendix A). The first and second axis accounted for 7% and 5% of the total variance, with eigenvalues of 0.69 and 0.46, respectively. The scores of each species calculated according to the left (sites) and right (traits) set were joined by a line in the figure. The species showing preferences for the cold sites at high altitude were plotted at the bottom right of the graphs, the ones present in sites with high oxygen content at the bottom left, tolerant species such as *Chironomus riparius* (= *thummi*), *Cricotopus (Cricotopus) trifascia* and *Virgatanytarsus* present in high N-NO_3_, TP, N-NH_4,_ and low oxygen content waters were plotted at the top part of the graph, *Rheopelopia*, *Uresipedilum*, *Tanypus* from sites with high temperature and conductivity were mapped at the top left part. An arch/horseshoe effect was also visible here, with species preferring lentic waters plotted on the left, kryal and cold spring species on the bottom right, and species characterizing potamon at the top right.

A comparison between factor loadings of species in canonical constrained and unconstrained ordination showed a good agreement in the species ordination, except for a few species such as *Diamesa dampfi*, *Micropsectra notescens*, *Paratrissocladius*, *Paracricotopus*, *Psectrocladius sordidellus*, *Heterotrissocladius*, which showed different scores in the CA first axis (calculated from **_n_L_p_** matrix) and in the CCA first axis (calculated from **_p_L’_n_**~**_p_U_s_** matrices) (Appendix A, Appendix A) and, as a consequence, were plotted at some distance from the regression line.

The **sites × species** matrix **_n_L_p_** was post-multiplied by **species × traits _p_U_s_** matrix to obtain a **site × traits** matrix (**_n_L _p_U_s_**). This **_n_L_p_U_s_** was also subjected to correspondence analysis (Figure 10, Appendix A). In this case, sites were rows and traits were columns. The first two axes accounted for 72% and 24% of the total variance, with eigenvalues of 0.05 and 0.02, respectively. The first axis reproduced an upstream-downstream and a water temperature gradient, and the second axis reproduced a water quality gradient (Figure 10). This analysis does not allow the mapping of species, because the species (columns of the first matrix and rows of the second) do not appear in the product matrix.

A discriminant analysis was carried out to test the goodness of classification of sites in different habitats when Chironomid taxa assemblages are used to discriminate among habitats (Table 4, Appendix A). Both **_n_L_p_** and **_n_L_p_U_s_** matrices were subjected to multiple discriminant analysis, using habitat as a grouping factor. Percent of correct classifications was 46% for the **_n_L_p_** matrix and 47% for the **_n_L _p_U_s_**, emphasizing that the addition of the trait matrix does not improve the classification significantly. In any case, the result is that Chironomid assemblages are good discriminators between the different habitats.

A cluster analysis of species confirmed that the separation of species clusters is in agreement with different habitats (Figure 11).

## 4. Discussion

Chironomid species distribution in the environment was confirmed to be related to ecological conditions. The distribution of Chironomids linked to biogeographic factors was never observed within the western Palearctic area, except for the species linked to glacial areas [36], so biogeographic factors are not considered in the present discussion.

Chironomids have been frequently used as indicators of past climatic change [37], while it is impossible to establish the occurrence of alien species [38], even if it is expected. Some species such as *Polypedilum nubifer* are probably invaders [39], but it is impossible to state if and when they reached the West Palearctic region. It is well known that Chironomid distribution is related to ecological factors, such as water temperature [40,41], so an extension or reduction of the home range of a species is expected in relation to global warming [42]. Considering that different factors (e.g., winds) are good carriers of Chironomid species, and even long geographical barriers can be overcome [43], it is clear that species distribution is a quite dynamic process and at present alien species among Chironomids are expected, but their identity cannot be established.

However, with the ecological niche being known, it is possible to transpose the information given by each species into information about its habitat. From a mathematical point of view, the ecological niche can be expressed as a vector whose elements are the optimum values of the species for each factor, expressed as a weighted mean, while the measure of niche extension can be expressed as a weighted standard deviation [31]. The vectors can be aggregated to create a trait matrix _p_U_s_ with p species as rows and s traits as columns. This _p_U_s_ matrix was firstly proposed for calculating aquatic beetle traits and a fuzzy coding analysis was suggested to allow the inclusion of diverse kinds of biological information [44]. Species abundances can be expressed as a matrix _n_L_p_ with n samples as rows and p species as columns. Matrix multiplication of the matrix _n_L_p_ by the _p_U_s_ matrix generates a _n_L_p_U_s_ product matrix, with n sites as rows and s traits as columns; this approach was proposed for vegetation studies [45], and it was used for invertebrates living in running waters [46] and extended to Chironomids [22,23,47]:**_n_M_s_** = **_n_L_p_** ∗ **_p_U_s_**

This approach allows us to transpose the information given by a species list into ecological traits, allowing the construction of an index of environmental quality. Attempts to create the _p_U_s_ matrix for Chironomids and other benthic invertebrates were a matter of many efforts [4,48,49,50], but the results were obviously dependent on the database used for calculations. In the present paper, we tried to develop a new trait matrix considering the largest database available from collections of larval samples from both lotic and lentic habitats. This can produce results in disagreement with the ones produced considering a database constructed using data from lotic or lentic habitats alone. Indeed, traits of chironomids were often assigned without well-founded Appendix A. For example, this was underlined in estimating the recovery of lakes after measures of restoration from acidification [51]. Significant differences were observed between traits developed for North American and European species [52] and between Scandinavian and Mediterranean species [53]. Lack of information may lead to apparently contradictory results. For example, hemoglobin content, tube building ability, feeding habit, voltinism, and body size of Chironomid larvae suggested that hemoglobin-rich species, with tube building capacity and short generation time, may be dominant in disturbed sites, while the reverse should be expected in less disturbed sites. However, this approach gave some unexpected results, such as the presence of: 1- hemoglobin-rich species in less disturbed sites; 2- species with long generation time in disturbed sites [47], and/or 3- small body-sized species in less disturbed habitats [24]. These apparently conflicting results were explained by positing that oxygen deficit was not the only factor determining disturbed conditions. It was posited that not all hemoglobin-rich species are tolerant to low oxygen levels, and also, within the same site, that different species at different depths may show variation in hemoglobin concentrations [14]. For example, species belonging to *Polypedilum* may be responsible for this conflicting result, because this hemoglobin-rich genus is often present in undisturbed sites, possibly due to the presence of small oxygen-poor microhabitats included in large oxygen-rich habitats. Chironomini genera (*Chironomus*, *Glyptotendipes*, *Polypedilum*, *Paratendipes*, *Microtendipes*, etc.) are all hemoglobin-rich [14], but they show very different responses to pollution. The same is true for body size: the large *Chironomus* and *Propsilocerus* often prevail in disturbed sites, while it is expected that the small body-sized traits prevail in disturbed sites [54].

Another attractive approach is the so-called 4th corner solution problem [55,56], where the sites × species matrix **_n_L_p,_** the species × traits matrix **_p_U_s_**, and the sites × environmental variables matrix **_n_R_q_** are combined to produce a **_q_D_s_** = **_q_R’_n_L_p_U_s_** matrix, which allows a comparison between an expected and an observed community [56]. In the present case, the **_n_R_q_** matrix presents a lot of missing data, so this analysis was not performed. Caution is suggested in using this matrix approach to evaluate the ecological status because incomplete information available about the ecology of single taxa can lead to misleading results or false representations. This approach could be useful in the future when more accurate information will be available about different Chironomid species.

In the present study, as in many others [4,46,56], it is evident that Chironomid species respond to a limited number of factors, so they can be ordered according to a few gradients. We preferred to start the analysis ordering taxa with an unconstrained ordination method [31], because environmental data supporting the description of sampled sites were incomplete. Moreover, it is well known that the presence-absence of a species is not bound to the instantaneous point water condition, rather it is the result of an integration of factors over a relatively long time period, information that cannot be given by physical-chemical analysis.

Despite these limitations, the ordination of sites, based only on Chironomid species assemblages available in the present database, emphasized few major gradients responsible of the observed distributions: 1- a gradient separating lotic from lentic habitats, with species living in fast-running waters separated from species living in standing waters; 2- a gradient emphasizing an upstream-downstream gradient in running waters, separating: (a) intolerant species living at high altitudes, low water temperatures, high oxygen concentrations, low conductivity, from (b) tolerant species living downstream, at higher temperatures, lower oxygen concentrations, higher conductivity, and salinity; 3- a trophic gradient separating species living in oligotrophic nutrient-poor waters from species living in organic-rich or eutrophic waters. Each of these gradients does not necessarily coincide with the principal axes resulting from canonical ordination. In the present case, the first axis separated lotic from lentic habitats, the second axis was explained as an oxygen-temperature gradient, and the ordering of sites resulted in the classic arch or horseshoe effect [31,32]. This effect observed in the correspondence analysis [31] is generated by species data having unimodal distribution along a single gradient [32]; in the present case, it was a gradient from high altitude, cold, oxygen-rich, fast-flowing running waters observed in glacial streams, toward lowland, warmer, oxygen-poor, slow-flowing waters observed in lowland rivers, and continuing in still slow flowing, but cooling down and oxygen enriching waters, as observed in large lakes with increasing depth. Conductivity and nutrients were often included in this principal gradient, in several possible interactions. In relation to this principal gradient, each species can adjust with its own peculiarities, moving more or less far from this gradient. For example, species living in small-sized cold waters lakes at high altitudes (*Zavrelimyia*, *Heterotrissocladius*, *Corynoneura*, *P. austriacus*) and species living at high depth in large lakes (*M. radialis*, *Paracladopelma*) were displaced toward the center of the plot (Figure 2 and Appendix A).

Species cannot be clustered in well-defined groups, because only a few species are restricted to well-defined habitats, while most species are opportunistic. For example, few species belonging to *Diamesa* are restricted to kryal (*Diamesa laticauda*), but most (*Diamesa tonsa*, *Diamesa zernyi*) colonize different types of cold waters; some Orthocladiinae genera (*Eukiefferiella*, *Rheocricotopus*, *Euorthocladius*, *Orthocladius*, *Cricotopus*) characterize rhithral streams with moderate or fast current, but can be collected also in slow flowing waters; many Tanytarsini are typical of oligotrophic lakes, but are also common in springs and streams; many Chironomini genera (e.g., *Dicrotendipes, Chironomus*) characterize eutrophic lakes, but many of them live also in potamal river stretches and in littoral zones of lakes associated to vegetation (*Endochironomus*, *Glyptotendipes*) or to sand banks (*Cryptochironomus*, *Harnischia*) [57].

In conclusion, the key factors separating Chironomid species are confirmed to be substrate type, current velocity, water temperature, dissolved oxygen, conductivity, and nutrients, but these factors are differently related in various situations and anthropogenic stress and can contribute to creating other more complex interactions [15].

The advantage of having a matrix of ecological traits available (_p_U_s_) is the possibility of using only taxonomic assemblage structure information to evaluate the ecological status of an ecosystem, without the support of environmental data. This is a necessity when sampling campaigns include only the monitoring of macrobenthos; in this case, if a trait matrix is available, taxonomic information can be transposed into water quality assessment.

## 5. Conclusions

It is often stated that functional traits analysis is better than taxonomic composition analysis [23]. Indeed, this statement stresses the obvious, because the use of functional traits requires having a trait matrix available, and the development of a trait matrix implies the existence of sound taxonomic knowledge, needed to create the trait matrix. It is more appropriate to state that when a trait matrix is available, less thorough taxonomic knowledge is needed to evaluate the ecological status of a water body. In other words, a species groups list, instead of a more thorough species list, can be sufficient to analyze the system. The traits matrix approach has the advantage that a taxonomic species list can provide information comparable with the one given by a physical-chemical analysis when a trait matrix is available. If both a traits matrix **_p_****U_s_** and an environmental variables matrix **_n_****R_q_** are available, a further step can be done, calculating an expected ecological status and comparing it with an observed one [58].

## Figures and Tables

**Figure 1 insects-13-00911-f001:**
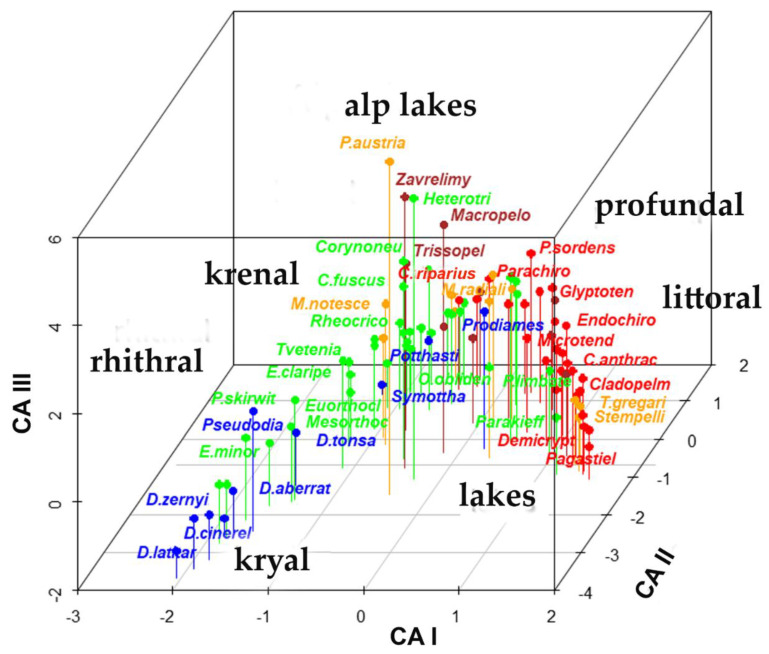
Plot of the species scores in the first 3 axes resulting from CA carried out from sites × species (**_n_L_p_**) matrix (the full set of species names is in Appendix A). Different subfamilies are plotted with different colors: brown: Tanypodinae; blue: Diamesinae and Prodiamesinae; green: Orthocladiinae; orange: Tanytarsini (tribe); red: Chironomini (tribe).

**Figure 2 insects-13-00911-f002:**
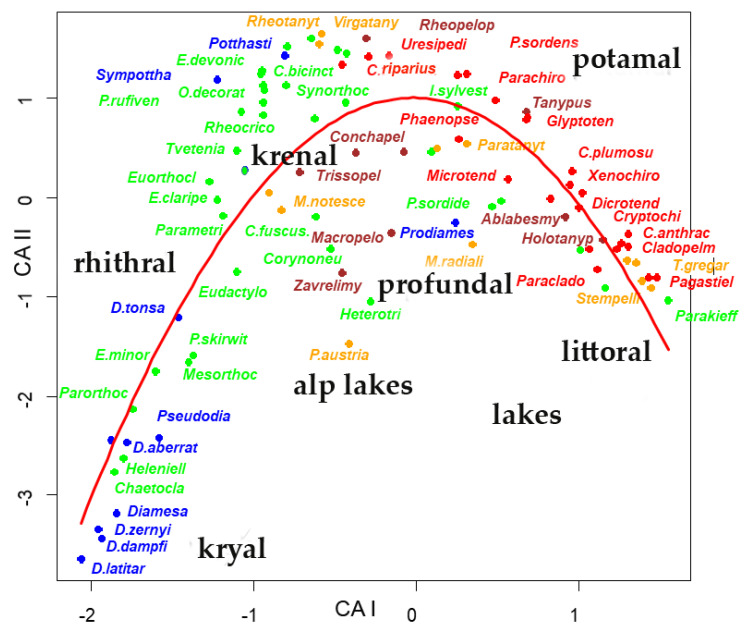
Plot of the species scores in the first 2 axes resulting from CA carried out from sites × species (**_n_L_p_**) matrix and the fitted second-degree polynomial (the full set of species names is in Appendix A). Different subfamilies are plotted with different colors: brown: Tanypodinae; blue: Diamesinae and Prodiamesinae; green: Orthocladiinae; orange: Tanytarsini (tribe); red: Chironomini (tribe).

**Figure 3 insects-13-00911-f003:**
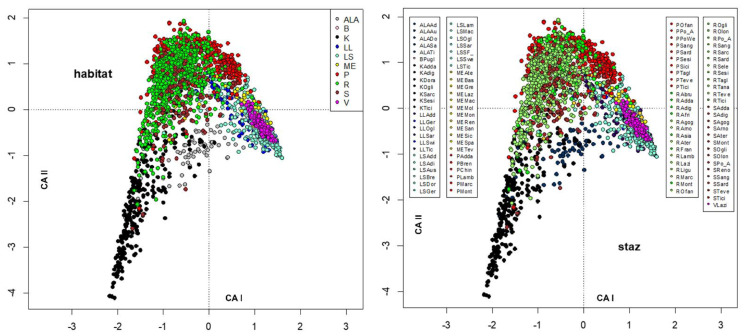
Plot of site scores in the first 2 axes resulting from CA of sites × species (**_n_L_p_**) matrix, by marking sites with different colors according to habitat (**left**) and to sampling station (**right**). In the right legend, the abbreviations of the sampling stations include habitat (see Section 2.1 and Appendix A) and the abbreviated name of the sampling station (see Appendix A).

**Figure 4 insects-13-00911-f004:**
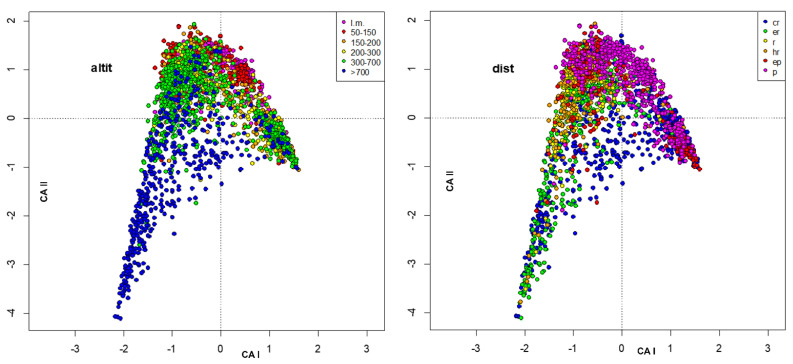
Plot of site scores in the first 2 axes resulting from CA of sites × species (**_n_L_p_**) matrix, by marking sites with different colors according to altitude (m a.s.l.) (**left**) and distance from the source (km) (**right**); abbreviations of right legend: cr: crenal, er: epirhithral, r: rhithral, hr: hyporhithral, ep: epipotamal, p: potamal.

**Figure 5 insects-13-00911-f005:**
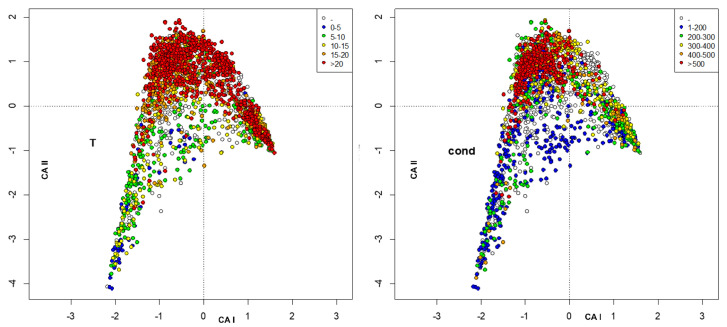
Plot of site scores in the first 2 axes resulting from CA of sites × species (**_n_L_p_**) matrix, by marking sites with different colors according to water temperature (°C) (**left**) and to water conductivity (µS cm^−1^) (**right**).

**Figure 6 insects-13-00911-f006:**
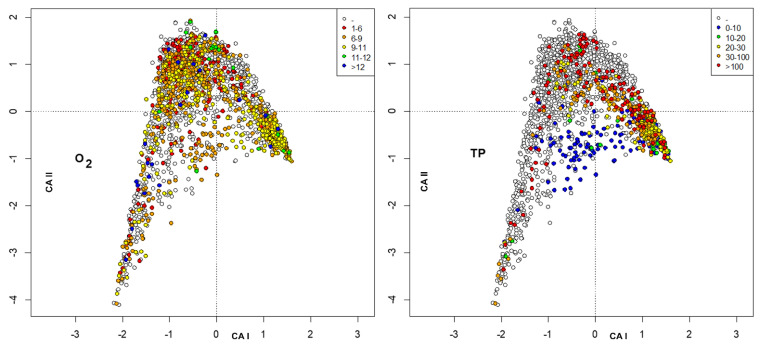
Plot of site scores in the first 2 axes resulting from CA of sites × species (**_n_L_p_**) matrix, by marking sites with different colors according to dissolved oxygen (mg L^−1^) (**left**) and total phosphorous (TP) (µg P L^−1^) (**right**).

**Figure 7 insects-13-00911-f007:**
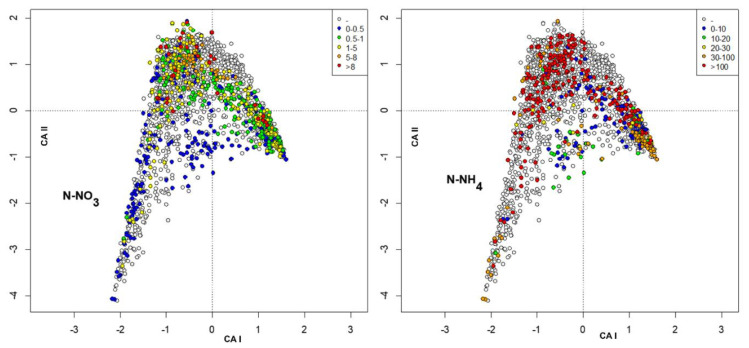
Plot of site scores in the first 2 axes resulting from CA of sites × species (**_n_L_p_**) matrix, by marking sites with different colors according to N-NO_3_ (mg N l^−1^) (**left**) and to N-NH_4_ (µg N l^−1^) (**right**).

**Figure 8 insects-13-00911-f008:**
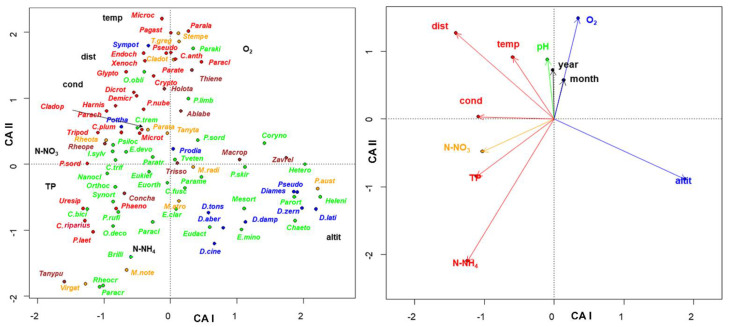
Plot of the species scores (**left**), and of the trait scores (**right**) in the first 2 axes resulting from CA carried out from species × traits (**_p_U_s_**) matrix. Different subfamilies are plotted with different colors: brown: Tanypodinae; blue: Diamesinae and Prodiamesinae; green: Orthocladiinae; orange: Tanytarsini (tribe); red: Chironomini (tribe). The different colors of environmental variables indicate their values increase with water quality (blue) or with water pollution (red), according to EQR colors [35].

**Figure 9 insects-13-00911-f009:**
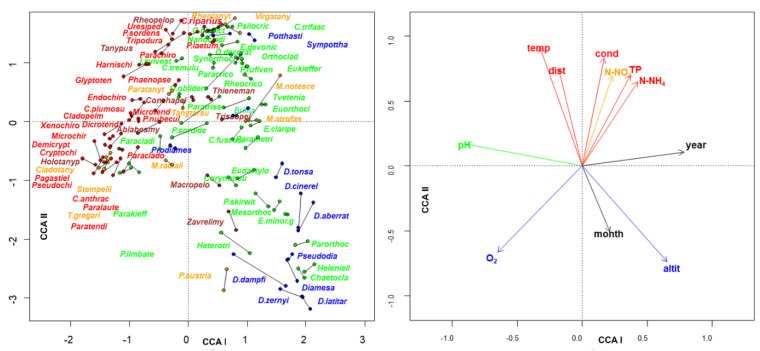
Plot of the species scores (**left**) of the trait scores (**right**) in the first 2 axes resulting from CCA analysis carried out from **_p_L’_n_**~**_p_U_s_** matrices; the scores of the same species obtained with the first and second matrix are joined with a line (see Appendix A for the full set of species names). Different subfamilies are plotted with different colors: brown: Tanypodinae; blue: Diamesinae and Prodiamesinae; green: Orthocladiinae; orange: Tanytarsini (tribe); red: Chironomini (tribe). The different colors of environmental variables indicate their values increase with water quality (blue) or with water pollution (red), according to EQR colors [35].

**Figure 10 insects-13-00911-f010:**
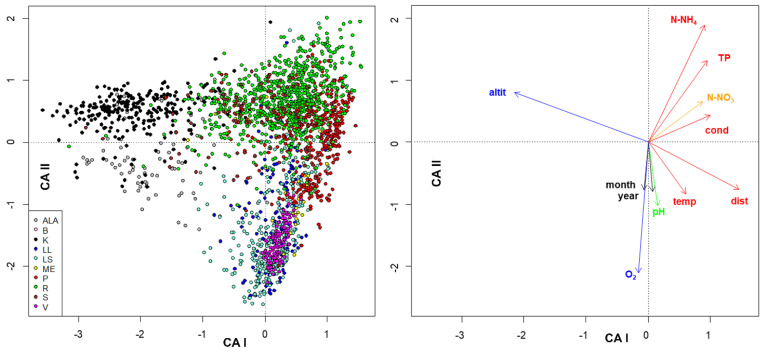
Site scores (**left**) and traits scores (**right**) from the first 2 axes of the site × traits (_n_L_p_U_s_) matrix. The abbreviations of the legend of the left figure are habitats (See Section 2.1 and Appendix A). The different colors of environmental variables in the right figure indicate their increase with water quality (blue) or with water pollution (red), according to EQR colors [35].

**Figure 11 insects-13-00911-f011:**
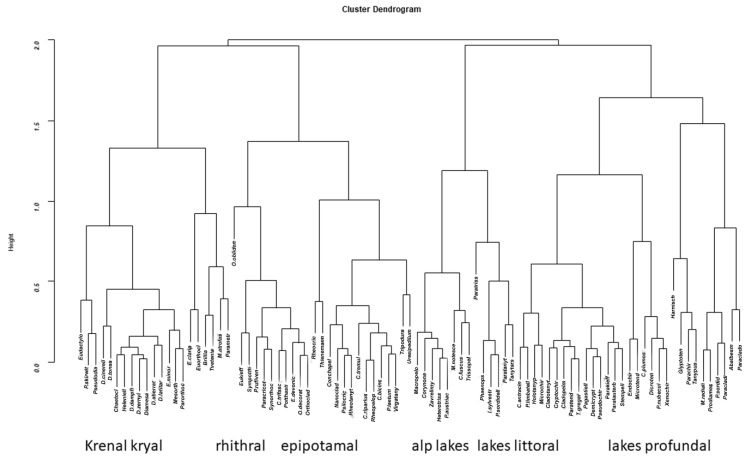
Cluster analysis of species from sites × species (_n_L_p_) matrix.

**Table 1 insects-13-00911-t001:** Number of sites available for each environmental variable: alt = altitude, dist = distance from the source, year, month, temp = water temperature, cond = conductivity, pH, O_2_ = dissolved oxygen, TP = total phosphorous, N-NO_3_ = nitrate nitrogen, N-NH_4_ = ammonium nitrogen.

Alt	Dist	Year	Month	Temp	pH	Cond	O_2_	N-NO_3_	TP	N-NH_4_
9127	7546	9127	9045	5951	4797	4823	5335	3530	2854	2045

**Table 2 insects-13-00911-t002:** Highly significant correlations (* = *p* < 0.01) between species and environmental variables, + = positive correlations, − =negative correlations. Detailed results in Appendix A. See Table 1 for abbreviations.

Species\Environmental Variables	Alt		Dist		Year		Month		Temp		pH		Cond		O_2_		N-NO_3_		TP		N-NH_4_	
*Ablabesmyia* spp.	+	*	−	*	+	*	+		−	*	+	*	−	*	+	*	−		−	*	−	
*Brillia* spp.	−		−		−		+		−		−	*	−	*	−		+		−		−	
*Chaetocladius* spp.	+	*	−	*	+	*	+		−		−	*	−		+		−		+		−	
*Chironomus (Chironomus) anthracinus*	+	*	−		+		+		−		+	*	−		+		−		−		−	
*Chironomus (Chironomus) plumosus*	+	*	−	*	−	*	−	*	−	*	+	*	−		+	*	+		−	*	−	
*Chironomus (Chironomus) riparius*	+		−		+		+		−		+	*	−		−		+		+		−	
*Cladopelma* spp.	+		−	*	+		+	*	+		+		−		+		−		−		−	
*Cladotanytarsus* spp.	+	*	−	*	−		−		−	*	+	*	−		+	*	+		−		−	*
*Conchapelopia* spp.	+		−		−	*	+		−	*	+	*	+		+	*	−		−	*	−	*
*Corynoneura* spp.	+	*	−		+	*	+	*	−	*	−		−	*	+		−	*	−	*	−	*
*Cricotopus (Cricotopus) bicinctus*	−		−		+		−		−	*	+		+		+	*	+		−		−	
*Cricotopus (Cricotopus) fuscus*	+	*	−		+		−		−	*	+		−		+		−		−		−	*
*Cricotopus (Cricotopus) tremulus*	−		−		−		−		−		−		+		+		−		−		−	
*Cricotopus (Cricotopus) trifascia*	−		+		−		−		−		+		+		+		+		−		−	
*Cricotopus (Isocladius) sylvestris*	−		−		−	*	−		−		+		−		+		−		−		−	
*Cricotopus (Paratrichocladius) rufiventris*	−		−		+	*	−		+		+		−		−		+		+		+	
*Cricotopus (Paratrichocladius) skirwithensis*	−		+		+		−		−		+		−		−		−		−		−	
*Cryptochironomus* spp.	+	*	−	*	−	*	−		−	*	+	*	−	*	+	*	−		−	*	−	*
*Demicryptochironomus* spp.	−		−		−	*	+		−		−		−		+		−		+		+	
*Diamesa aberrata*	+		+		+		+		−		−		+		+		+		−		−	
*Diamesa cinerella*	−		+		+		−		+		+		−		−		−		−		−	
*Diamesa dampfi*	+		−		+		+		−		+		−		+		−		−		−	
*Diamesa latitarsis*	+	*	−	*	+	*	+		−	*	−	*	−	*	+		−		−		−	
*Diamesa* spp.	+	*	−		+	*	+		−	*	−		−	*	+		−		−		−	
*Diamesa tonsa*	+	*	−		+		+		−	*	−	*	−	*	+		−		−		−	*
*Diamesa zernyi*	+	*	−		+	*	+		−	*	−	*	−	*	+		−	*	−		−	
*Dicrotendipes* spp.	+	*	−	*	+		−		−	*	−		−		+	*	−	*	−	*	−	*
*Endochironomus* spp.	+	*	−	*	−		+		−		+		−		+		−		−	*	−	
*Eukiefferiella claripennis*	+	*	−	*	+		−		−		−	*	−		−		−		−		−	
*Eukiefferiella devonica*	−		+		+	*	+		+		+		−		+		−		+		−	
*Eukiefferiella minor*	+	*	−		+	*	+		−	*	−	*	−	*	−		−	*	−		−	
*Eukiefferiella* spp.	+		+		+		−		+		+		−		−		+		−		−	
*Glyptotendipes* spp.	+		−		+		+		−		+	*	−	*	−		−		−		−	
*Harnischia* spp.	+	*	−	*	+	*	−		−	*	+	*	−		+	*	−		−	*	−	
*Heleniella* spp.	+	*	−		+	*	+		−		−		−	*	+		−		−		−	
*Heterotrissocladius marcidus*	+	*	−		+	*	+	*	−	*	−		−	*	+	*	−	*	−		−	*
*Macropelopia* spp.	+	*	+		+	*	+		−	*	−		−	*	+		−		−		−	
*Microchironomus* spp.	−		+		−		−		−		+		−		+	*	−		−		−	*
*Micropsectra atrofasciata*	+	*	−	*	+	*	−		−	*	−	*	−	*	−		−	*	−	*	−	
*Micropsectra notescens*	+		+		+		−		+		+		+		−		+		+		+	
*Micropsectra radialis*	+		+		−		−		−		+		−	*	+	*	−		−		−	
*Microtendipes* spp.	+		−		−	*	−		−		+	*	−		+	*	−		−		−	
*Nanocladius* spp.	−		−		+		+		+		−	*	+		−		−		−		−	
*Orthocladius (Eudactylocladius)* spp.	+		+	*	+	*	−		−		−		−		+		−		−		−	
*Orthocladius (Euorthocladius)* spp.	+	*	−	*	+	*	−		−		−	*	+		−		−		−	*	−	*
*Orthocladius (Mesorthocladius)* spp.	+	*	−		+	*	+		−	*	−		−	*	−		−		−		−	
*Orthocladius (Orthocladius) decoratus*	+		−	*	+	*	−		−		−		+		+		+		−		−	
*Orthocladius (Orthocladius) oblidens*	+		−		−	*	−		−	*	−		−		+		−		−	*	−	
*Orthocladius (Orthocladius)* spp.	−		−		+	*	−	*	+		+		+		−		−		−		−	
*Pagastiella orophila*	+		−	*	+		−		−		+		+	*	+		−		−		−	
*Parachironomus* spp.	−		−		−	*	−		−		+		+		+		+		−		−	
*Paracladius* spp.	+		−		+		+		−		+		−		+		−		−		+	
*Paracladopelma* spp.	+	*	−	*	+	*	−	*	−		+		−		+		+	*	+		−	
*Paracricotopus niger*	−		−		+		−		+		−		+		−		−		+		+	
*Parakiefferiella* spp.	−		+		+		−		+		+		−		+	*	+		−		−	
*Paralauterborniella nigrohalteralis*	+	*	−	*	+	*	−		−	*	+	*	+		+		+		−		−	
*Parametriocnemus* spp.	+	*	−		+	*	+		−	*	−	*	−		+		−		−		−	
*Paratanytarsus austriacus*	+		−		+		+		−		+		−		+		−		−		+	
*Paratanytarsus* spp.	+	*	−	*	−		−		−		+		−		+	*	−		−		−	
*Paratendipes* spp.	−	*	−	*	−	*	−	*	−		+	*	−		+	*	−		−	*	−	
*Paratrissocladius excerptus*	+		+		+		−		+		−		+		−		+		−		−	
*Parorthocladius nudipennis*	+	*	−		+	*	+		−	*	−		−	*	+		−		−		−	
*Phaenopsectra flavipes*	+	*	−		+		−		−		+		−		+	*	−		−	*	−	
*Polypedilum (Pentapedilum) sordens*	−		+		−		−		−		+		+		−		−		−		+	
*Polypedilum (Polypedilum) laetum*	+		−		+		+		−		+		+		+		+		−		−	
*Polypedilum (Polypedilum) nubeculosum*	+		−	*	−	*	−		−		+	*	−	*	+	*	−		−		−	*
*Polypedilum (Tripodura)* spp.	+	*	−	*	+	*	+		−		+		+		+		−		−		−	
*Polypedilum (Uresipedilum)* spp.	+		−		−		−		−		+		+		+		−		−		−	
*Potthastia* spp.	−		−		+	*	−		+		+		−		−		−		−		−	
*Procladius (Holotanypus) choreus*	+		−	*	−		−		−	*	+	*	−	*	+	*	−		−	*	−	*
*Prodiamesa* spp.	+		+	*	+		+		−	*	+	*	−	*	+		−		−		−	
*Psectrocladius (Psectrocladius) limbatellus*	+		−	*	−	*	−		−	*	+		−	*	+	*	−		−		−	
*Psectrocladius (Psectrocladius) sordidellus*	+		−		+		+		−		+		−		+		−	*	−		−	
*Pseudochironomus prasinatus*	+		−	*	−		−		−		+		−		+	*	−		−	*	−	
*Pseudodiamesa* spp.	+	*	+		+	*	+		−	*	−	*	−	*	−		−		−		−	
*Rheocricotopus (Psilocricotopus)* spp.	−		−		+		+		+		+		+		−		−		−		−	
*Rheocricotopus (Rheocricotopus)* spp.	−		−		+		−		+		+		−		−		−		+		+	
*Rheopelopia ornate*	+		+	*	+		+		−	*	+		−		−		+		−		−	
*Rheotanytarsus* spp.	+		−		+		+		−		+		−		+		−		+		−	
*Stempellina bausei*	+	*	−	*	+		+		−		+		+		+	*	+		+		−	
*Sympotthastia spinifera*	+		−		+	*	−		−		+		−		−		−		−		−	
*Synorthocladius semivirens*	+	*	−		+	*	−		−		−		−		−		−		−		−	
*Tanypus (Tanypus) punctipennis*	−		+		−		+		+		+		−		−		−		+		+	
*Tanytarsus gregarious*	+	*	−	*	+		+		−		−		−		+	*	−		−		−	*
*Tanytarsus* spp.	+	*	−	*	+	*	−		−	*	+	*	−		+	*	−		−	*	−	*
*Thienemannimyia* spp.	+	*	−		+		+		−		+		−		+		−		−		−	
*Trissopelopia longimanus*	+		+		−	*	−		−		−		−		−		−		−		−	
*Tvetenia* spp.	+		+		+	*	−		+		+		−		−		−		−		−	
*Virgatanytarsus* spp.	+		−		+		+		+		+		+		−		−		−		−	
*Xenochironomus xenolabis*	+		−	*	−	*	−		−		+	*	−		+	*	−		−		−	
*Zavrelimyia* spp.	+	*	−		+	*	+		−	*	−		−		+		−		−		−	

**Table 3 insects-13-00911-t003:** Matrix of traits: weighted mean of each environmental variable for each species. Standard deviations and number of sites used in Appendix A.

Species\Traits	Alt	Dist	Year	Month	Temp	pH	Cond	O_2_	N-NO_3_	TP	N-NH_4_
*Ablabesmyia* spp.	516	41	1999	6	18	7	273	7	1	69	239
*Brillia* spp.	522	19	1988	6	17	7	328	5	2	207	973
*Chaetocladius* spp.	1872	4	1996	8	8	6	75	7	1	80	241
*Chironomus (Chironomus) anthracinus*	319	55	1981	6	17	8	239	8	1	53	119
*Chironomus (Chironomus) plumosus*	212	95	1981	6	20	7	349	6	1	127	469
*Chironomus (Chironomus) riparius*	223	122	1993	6	22	7	669	4	2	335	920
*Cladopelma* spp.	284	40	1983	7	19	7	281	7	1	98	410
*Cladotanytarsus* spp.	361	24	1989	6	17	8	318	8	1	57	101
*Conchapelopia* spp.	414	53	1987	6	18	7	632	5	2	99	696
*Corynoneura* spp.	1229	33	1995	7	15	7	180	6	0	26	84
*Cricotopus (Cricotopus) bicinctus*	220	109	1993	6	21	7	775	5	3	212	885
*Cricotopus (Cricotopus) fuscus*	809	28	1992	6	16	7	415	4	2	34	502
*Cricotopus (Cricotopus) tremulus*	354	94	1990	6	20	7	455	4	1	143	372
*Cricotopus (Cricotopus) trifascia*	269	65	1996	6	21	7	569	3	3	141	560
*Cricotopus (Isocladius) sylvestris*	280	118	1990	6	22	7	649	4	1	126	543
*Cricotopus (Paratrichocladius) rufiventris*	432	42	1996	6	20	7	678	4	2	176	778
*Cricotopus (Paratrichocladius) skirwithensis*	1193	20	1994	7	11	7	249	5	1	70	200
*Cryptochironomus* spp.	269	54	1982	6	18	7	281	8	1	67	218
*Demicryptochironomus* spp.	217	62	1977	7	20	7	314	8	1	116	298
*Diamesa aberrata*	1329	15	1985	6	12	7	279	4	2	246	457
*Diamesa cinerella*	1272	35	1998	6	11	7	218	4	1	178	626
*Diamesa dampfi*	1384	7	1990	6	11	7	147	8	1	114	428
*Diamesa latitarsis*	2089	4	1999	8	7	6	108	8	1	61	90
*Diamesa* spp.	1735	6	1992	8	8	6	158	6	0	50	118
*Diamesa tonsa*	1051	22	1991	6	15	7	253	5	2	124	509
*Diamesa zernyi*	1917	6	1996	8	8	6	127	7	0	53	153
*Dicrotendipes* spp.	254	74	1987	6	20	7	513	6	1	72	278
*Endochironomus* spp.	215	90	1981	7	19	8	352	4	1	64	165
*Eukiefferiella claripennis*	830	33	1994	6	17	7	433	5	1	101	606
*Eukiefferiella devonica*	305	79	1998	6	21	7	441	4	2	154	505
*Eukiefferiella minor*	1433	15	1993	7	11	7	219	5	1	76	486
*Eukiefferiella* spp.	511	81	2000	6	19	7	352	3	1	190	509
*Glyptotendipes* spp.	164	176	1985	7	21	7	264	4	1	86	297
*Harnischia* spp.	205	155	1991	6	21	8	535	5	1	73	397
*Heleniella* spp.	1978	6	1998	8	8	6	62	7	0	29	59
*Heterotrissocladius marcidus*	1604	14	1995	7	11	7	69	7	0	13	28
*Macropelopia* spp.	1081	32	1998	7	13	7	180	7	1	61	219
*Microchironomus* spp.	233	64	1988	6	16	8	296	7	1	38	49
*Micropsectra atrofasciata*	807	35	1994	6	17	7	382	4	1	105	559
*Micropsectra notescens*	651	45	1989	6	19	7	600	2	4	180	1107
*Micropsectra radialis*	743	62	1996	7	16	7	187	8	1	81	414
*Microtendipes* spp.	341	48	1990	6	20	7	459	6	2	80	409
*Nanocladius* spp.	252	108	1990	7	24	7	497	4	1	267	616
*Orthocladius (Eudactylocladius)* spp.	1144	39	1995	7	15	7	248	6	1	100	596
*Orthocladius (Euorthocladius)* spp.	702	59	1994	6	18	7	394	4	1	218	467
*Orthocladius (Mesorthocladius)* spp.	1354	16	1994	6	13	7	224	5	1	64	381
*Orthocladius (Orthocladius) decoratus*	425	107	1997	5	21	7	528	5	2	242	886
*Orthocladius (Orthocladius) oblidens*	261	66	1987	5	19	7	422	6	1	87	186
*Orthocladius (Orthocladius)* spp.	331	69	1994	5	20	7	560	4	2	199	663
*Pagastiella orophila*	294	50	1979	6	20	7	263	9	1	60	54
*Parachironomus* spp.	136	143	1986	6	23	7	466	4	2	161	409
*Paracladius spp.*	607	54	1992	6	17	7	291	6	1	105	772
*Paracladopelma* spp.	530	54	1997	5	17	8	213	8	1	57	50
*Paracricotopus niger*	391	38	1996	6	19	7	506	5	1	225	1292
*Parakiefferiella* spp.	426	36	1987	5	16	7	208	9	1	47	51
*Paralauterborniella nigrohalteralis*	378	48	1992	6	13	7	236	9	1	16	31
*Parametriocnemus* spp.	896	19	1994	6	15	7	360	5	1	73	382
*Paratanytarsus austriacus*	1899	3	2000	8	10	7	81	8	0	5	51
*Paratanytarsus* spp.	408	85	1988	6	21	7	427	5	2	60	391
*Paratendipes* spp.	370	33	1979	6	16	8	308	7	1	45	102
*Paratrissocladius excerptus*	535	17	1996	6	16	7	623	7	1	75	428
*Parorthocladius nudipennis*	1806	4	1997	8	10	6	187	6	0	55	122
*Phaenopsectra flavipes*	335	69	1995	6	21	7	450	6	1	127	812
*Polypedilum (Pentapedilum) sordens*	123	179	1990	6	22	7	596	2	1	116	718
*Polypedilum (Polypedilum) laetum*	312	111	1990	6	23	7	681	4	3	283	974
*Polypedilum (Polypedilum) nubeculosum*	302	72	1983	6	19	7	357	6	1	78	334
*Polypedilum (Tripodura) spp.*	214	156	1995	7	23	7	853	5	1	202	459
*Polypedilum (Uresipedilum)* spp.	249	129	1990	7	22	7	998	5	4	191	909
*Potthastia* spp.	211	107	1996	5	22	7	349	4	2	140	448
*Procladius (Holotanypus) choreus*	367	48	1985	6	17	7	300	8	1	67	216
*Prodiamesa* spp.	557	53	1990	6	16	7	263	7	1	92	389
*Psectrocladius (Psectrocladius) limbatellus*	497	29	1980	6	17	7	207	7	1	66	194
*Psectrocladius (Psectrocladius) sordidellus*	693	35	2001	6	19	7	214	6	1	92	283
*Pseudochironomus prasinatus*	290	36	1981	6	20	7	273	8	1	55	123
*Pseudodiamesa* spp.	1733	16	1993	7	9	7	109	5	0	28	135
*Rheocricotopus (Psilocricotopus)* spp.	237	90	1994	7	22	7	503	4	2	265	457
*Rheocricotopus (Rheocricotopus)* spp.	501	46	1992	6	18	7	698	4	3	225	1300
*Rheopelopia ornata*	226	249	1993	6	18	7	395	5	3	201	573
*Rheotanytarsus* spp.	225	152	1992	6	20	7	452	4	2	456	393
*Stempellina bausei*	322	51	1984	6	16	7	215	9	1	55	52
*Sympotthastia spinifera*	246	88	2003	4	21	7	365	4	1	122	94
*Synorthocladius semivirens*	313	81	1996	6	21	7	444	4	1	188	764
*Tanypus (Tanypus) punctipennis*	170	133	1990	7	24	7	763	4	2	283	1448
*Tanytarsus gregarius*	339	50	1981	6	17	7	228	8	1	42	74
*Tanytarsus* spp.	573	65	1996	6	19	7	480	6	2	67	320
*Thienemannimyia* spp.	599	55	2002	6	17	7	492	6	1	16	38
*Trissopelopia longimanus*	680	38	1992	7	16	7	438	3	0	22	428
*Tvetenia* spp.	668	38	1995	6	18	7	429	5	2	93	387
*Virgatanytarsus* spp.	375	49	1995	7	22	7	951	3	5	114	1417
*Xenochironomus xenolabis*	224	56	1976	6	20	7	387	7	1	59	174
*Zavrelimyia* spp.	1480	11	1997	7	12	7	176	7	0	29	41

**Table 4 insects-13-00911-t004:** Results of discriminant analysis: hits and misses in samples classification according to taxonomic and traits analysis. ALA: alpine lakes, B: brackish waters, K: kryal, LL: large lakes, LS: small lakes, ME: Mediterranean lakes, P: potamal, R: rhithral, S: krenal, V: volcanic lakes. Detailed results of Discriminant Analysis are provided in Appendix A.

		ALA	B	K	LL	LS	ME	P	R	S	V
** _n_ ** **L_p_**	hits	56	100	79	9	11	28	40	38	33	68
	misses	44	0	21	91	89	72	60	62	68	32
** _n_ ** **L_p_U_s_**	Hits	59	100	83	14	10	28	36	34	38	72
	misses	41	0	17	86	90	72	64	66	63	28

## Data Availability

All specimens analyzed and the related data and all the figures produced are deposited at the University of Milan and can be requested to the first author.

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
