# Peer review of "Response of Chironomids to Key Environmental Factors: Perspective for Biomonitoring"

_insects, 2022, doi:10.3390/insects13100911_

Round 1

Reviewer 1 Report

Dear authors,

This is a good article. It is well-written and requires minor revisions. I have provided my detailed comments in the attached file.

Overall this article can be improved by:

1.       Increasing the number of relevant cited studies.

2.       Providing appropriate legends for your graph

3.       An explanation for the significance of environmental variables used in methodology

4.       Providing the sources of your data (cite)

5.       You are studying specific lentic and lotic habitats. This point must be made clear from the beginning so that your readership knows that you are not just looking for any lotic or lentic freshwaters.

6.       Few paragraphs and points made in the discussion lack appropriate citations. Please provide relevant references were needed.

Author Response

I have  joined the two answers to reviewers in only one file 

Reviewer 2 Report

Dear Authors,

This manuscript represents important research examining the species-environment relationships and traits matrices for an important and informative group of macroinvertebrates, Chironomidae. You make good use of the extensive database in your analyses, although you do not address possible biases arising from different collection methodologies.

I suggest that the paper be revised considerably before publication to address:

Errors in English language and cut and paste errors (see last line of the Conclusions).

Omission of analytical software used in the analyses.

A need to streamline your analyses and results to better meet the aim of your paper. 

I strongly encourage you to revise the paper and resubmit.

Best wishes

Author Response

I have uploaded the response to the two reviewers in only one file
